# Laryngeal Cancer Surgery: History and Current Indications of Transoral Laser Microsurgery and Transoral Robotic Surgery

**DOI:** 10.3390/jcm11195769

**Published:** 2022-09-29

**Authors:** Stéphane Hans, Robin Baudouin, Marta P. Circiu, Florent Couineau, Quentin Lisan, Lise Crevier-Buchman, Jerome R. Lechien

**Affiliations:** 1Department of Otorhinolaryngology and Head and Neck Surgery, Foch Hospital, School of Medicine, UFR Simone Veil, Université Versailles Saint-Quentin-en-Yvelines (Paris Saclay University), 92150 Paris, France; 2Division of Laryngology and Broncho-Esophagology, EpiCURA Hospital, UMONS Research Institute for Health Sciences and Technology, University of Mons (UMons), 7000 Mons, Belgium; 3Polyclinic of Poitiers—Elsan, 86000 Poitiers, France

**Keywords:** laryngeal cancer, history, minimally invasive surgery, transoral laser microsurgery, transoral robotic surgery

## Abstract

The development of transoral laser microsurgery (TLM) was an important step in the history of conservative laryngeal surgery. TLM reported comparable oncological outcomes and better functional postoperative and rehabilitation outcomes than open partial laryngectomy. TLM is currently considered as the standard surgical approach for early-stage laryngeal carcinoma. However, TLM has many limitations, including the limited view of the surgical field through the laryngoscope, exposure difficulties for some tumor locations, and a long learning curve. The development of transoral robotic surgery (TORS) appears to be an important issue to overcome these limitations. The current robotic technologies used in surgery benefited from the research of the U.S. Military and National Aeronautics and Space Administration (NASA) in the 1970s and 1980s. The first application in humans started in the 2000s with the first robotic-assisted cholecystectomy in the US, performed by a surgeon located in France. The use of robots in otolaryngology occurred after the development of the *Da Vinci* system in digestive surgery, urology, and gynecology, and mainly concerns cT1-T2 and some selected cT3 oropharyngeal and supraglottic carcinomas. With the development of a new robotic system with smaller arms and instruments, TORS indications will probably evolve in the next few years, leading to better outcomes for laryngeal or hypopharyngeal carcinomas.

## 1. Introduction

The treatment of laryngeal cancer through partial laryngectomy evolved over the past centuries. The second part of the 20th century was characterized by the development of fiberoptic, endoscopic, and robotic technologies, leading to a progressive evolution of practices from open partial laryngeal surgery to transoral laser microsurgery (TLM) and, recently, transoral robotic surgery (TORS). The objective of TLM and TORS approaches are to reduce both morbidity and mortality related to postoperative complications while maintaining adequate overall and recurrence-free survival [1,2].

In this paper, we described the evolution of techniques, indications, and surgical outcomes of TLM and TORS partial laryngectomies.

## 2. Transoral Laser Microsurgery

### 2.1. The Beginning: Albert Einstein and Colleagues

The work of Albert Einstein (1879–1955) on the principle of stimulated emission (1917) was the basis of the development of laser in surgery. On 16 May 1960, at Hughes Malibu laboratories (Los Angeles, CA, USA), Maiman’s solid-state pink ruby laser emitted mankind’s first coherent light, with rays all the same wavelength and fully in phase. After the industrial applications of lasers, many applications were developed in the medical field. In 1972, Strong et Jako reported the first application of the CO_2_ laser in transoral microsurgery [3] for the resection of early-stage vocal cord cancer. The use of CO_2_ for TLM was spread from this period worldwide, reporting high rates of local control (90–94%) and low rates of ‘salvage’ total laryngectomy (0–4%) in selected T1a vocal fold carcinoma [4,5,6]. In 1992 and 1993, Eckel et Thumfart [7] and Steiner [8] were the first laryngologists to report that CO_2_ laser may be used for more extensive laryngeal carcinomas including glottic and supraglottic cT2-T3 cancers. At this time, the concept of Transoral CO_2_ Laser Microsurgery was definitively born.

### 2.2. Concept of TLM

The concept of TLM differs from the open partial surgery approach. In open partial laryngectomy, the resection is carried out through a systematic way, removing some defined anatomical structures according to a standard well-reported approach (for example: systematic resection of the thyroid cartilage in supracricoid laryngectomy). In TLM, the resection is based on the tumor location and invasion. Surgeon removes the abnormal tissue with margins and preserves normal structures to ensure adequate functional outcomes.

In 2000, the European Laryngological Society (ELS) published a classification of the different cordectomies that are classified according to six types (Figure 1) [9,10]. The ELS classification for TLM is widely used and considers both depth and surface area of the resection. Types I and II consist of subepithelial and subligamental resections, respectively. Type III cordectomy consists of transmuscular resection and may involve partial resection of the ventricular fold to obtain an adequate exposure of the entire vocal fold (hatched area in Figure 1). The entire vocal cord is excised in type IV cordectomy; the vocal fold access requesting partial or total ipsilateral ventricular fold resection (hatched area in Figure 1). The Va, b, c, and d types consist of extended cordectomies encompassing the contralateral vocal fold (a), the arytenoid (b), ventricle (c), or subglottis (d), respectively. In type VI, the anterior commissure of the vocal folds is excised.

### 2.3. Indications and Limitations of TLM

TLM are mainly indicated for early-stage vocal fold cancers (Tis, T1, and T2). Current limitations for TLM resection of glottic lesions were summarized by Peretti et al. [11] and included tumor with reduced vocal fold moving due to cricoarytenoid join fixation/infiltration. Of note, restricted mobility due to limited muscle infiltration or mass effect of tumor may be not considered as a contraindication. The involvement of the posterior paraglottic region and cricoarytenoid joint infiltration negatively influence oncological and functional outcomes, and the role of TLM is, therefore, limited to some selected and rare cases. The cricoid cartilage resection is a contraindication for TLM according to the risk of laryngeal stenosis and tracheotomy. In addition, surgeons have to keep in mind that at least one cricoarytenoid unit (cricoid, arytenoid, cricoarytenoid joint/muscles, and associated recurrent nerve) must be preserved to have adequate postoperative functional outcomes.

The size of the tumor (cT), the vocal fold fixation, the arytenoid invasion, the anterior commissure involvement, and the lateral extension to the lateral ventricle were identified as poor oncological outcomes [12].

The development of the TLM approach for vocal fold cancers led to the consideration of this endoscopic approach for supraglottic and hypopharyngeal tumors at the same time. A classification has been proposed by the European Laryngological Society [13]. It has been rapidly demonstrated that TLM was associated with similar oncological outcomes but better functional outcomes than open partial laryngopharyngeal surgeries for early and middle stages supraglottic and hypopharyngeal cancers The improved functional outcomes were related to the lack of tracheotomy, the faster rehabilitation of swallowing, and the shorter hospital stay [14,15,16]. Through the last decades, TLM became the first-line minimally invasive surgical approach for most early and moderate stages of the laryngeal and selected hypopharyngeal cancers. However, TLM is associated with a long learning curve [17] and technical difficulties, which support the need of experienced multidisciplinary teams. Interestingly, some teams used endoscopic approaches for cT3 and cT4 laryngeal cancers and reported similar oncological outcomes than those of total laryngectomy [16].

### 2.4. Benefits of TLM

TLM became the gold standard for surgical treatment of early vocal fold cancers through the two last decades because this approach was associated with many benefits.

First, TLM is associated with comparable or superior oncological outcomes, including high disease-free survival, local, or regional control rates, and larynx preservation rate, over partial surgery, or radiotherapy. According to large cohort studies, the local control rate of early glottic cancers is >90% and the laryngeal preservation rate is >95% [2,12,17,18,19]. Interestingly, Ambrosch and Fazel reported in a large cohort series that 5-year local control and laryngeal preservation rates of patients treated with TLM for Tis-T2 laryngeal carcinomas were higher than those who underwent radiotherapy [20]. Similar findings were reported in a recent meta-analysis that reported better overall survival, disease-free survival, and preservation rates of TLM over radiotherapy [21]. Concerning the comparison between open partial versus endoscopic approaches for cT1 vocal fold carcinoma, Karatzanis et al. reported similar oncological outcomes between both approaches but superiority of TLM over open surgery about postoperative morbidity outcomes [22]. The better short to long-term postoperative and quality-of-life outcomes of TLM *over* open partial surgery or radiotherapy were supported in additional studies [23]. Second, the healthcare system cost of TLM was significantly lower than those of partial laryngectomies or radiotherapy [24]. Thus, in Spain, the TLM treatment for cT1 vocal fold cancer costs EUR 2200, while it costs EUR 4800 and EUR 13,000 for radiotherapy or partial laryngeal surgery [24]. The lack of tracheotomy, feeding tube, and the related reduction in hospital stay, as well as the simplest follow-up of TLM patients, are additional advantages in term of cost. These advantages supported the fact that 96% of patients themselves prefer being treated by TLM rather than radiotherapy [25]. The aim of this study was to investigate treatment preferences in patients with early glottic carcinoma who were given a choice between TLM and RT. Patients were informed concerning each treatment by a laryngologist and a radiotherapist. Of one hundred and seventy-five patients, ninety-six percent chose TLM, and seven patients chose RT. The decision was based mainly on treatment duration and the wider treatment options in case of recurrence [25].

Third, another strength of TLM is the possibility to keep partial laryngectomy or radiotherapy for a second-line therapeutic approach in case of recurrence or second localization [26].

### 2.5. Barriers to TLM

#### 2.5.1. Learning Curve

The main publications demonstrating adequate outcomes of TLM are large cohort series, while studies with low numbers of patients showed less satisfactory results [26]. The learning curve of TLM is an important issue. In open partial surgery, the senior surgeon may easily learn and control the surgery steps carried out by a fellow or a resident. In TLM, this is more complicated because the surgical field view is better for the junior than for the senior surgeon. Thus, the TLM learning curve is commonly recognized as long [27], supporting the development of anatomical larynx models for the training.

#### 2.5.2. Exposure of the Larynx

Larynx exposure is an important issue to perform tumor TLM complete resection. Laryngeal exposure is a classical factor that may limit the realization of TLM in proportions varying from 1.5% up to 24.0% of cases, according to the current literature [28]. To date, only a few studies described the key factors associated with adequate glottic exposure [2,28,29]. Some technical points are important to know by otolaryngologists to improve the exposure of the glottic region. In our series of one hundred and forty-eight patients with early-stage glottic squamous cell carcinoma (Tis, T1, and T2), the exposure was adequately performed in one hundred and forty-one patients (95.3%) through external counter pressure (65.9%), partial (57.4%), or total vestibulectomy (4.2%) of cases, respectively. Seven patients (4.7%) were treated by open partial laryngectomy because of exposure limitation [2].

#### 2.5.3. Anterior Commissure Involvement

The involvement of the anterior commissure is a contributing factor of recurrence and is associated with lower local control, larynx preservation rate, and disease-specific survival [17,30]. However, the overall survival of early-stage laryngeal cancer with anterior commissure invasion was similar than those of laryngeal cancer without involvement of the anterior commissure [17]. The type of tumor extension (horizontal vs. vertical) is an additional important factor because vertical extension tumors may be more frequently associated with pre-epiglottic space or subglottic invasion compared with horizontal extension tumors, which is an additional poor prognostic factor [31]. The oncological results of this kind of tumor may be influenced by the experience of the surgeons [32].

#### 2.5.4. Resection Margins

It is broadly accepted that a proportion of patients with positive margins do not develop recurrence, and the relationship between resection margin status and recurrence rate is still unclear [30]. The pathological examination of post-TLM specimen is a challenging task following laser surgery inducing difficulties in margin status outcome. In case of suspected positive margins, three attitudes are reported in the literature. First, the patient will follow a TLM reintervention or radiotherapy. Second, a second endoscopic evaluation 6 to 8 weeks after the TLM may be performed to determine the need for reintervention. Third, the surgeon can propose a follow-up approach and a particular attention to voice quality impairment or videolaryngostroboscopy suspicion of recurrence/progression during close follow-up. The margin status did not seem to have negative consequences on the 5-year overall survival, disease-free survival, and laryngeal preservation rate of patients treated by experienced teams [2,17,31]. In that way, in 2014, the European Laryngological Society guidelines supported follow-up and re-evaluation for laryngeal cancer positive margins [33]. In 2019, a National Cancer Data Base study compared the overall survival rate of patients with positive margins with those with negative margins after TLM for glottic cT1-T2 cancers. Among the 747 patients treated between 2004 and 2013, 598 individuals (80.1%) had negative margins and the median follow-up time was 48 months. Authors reported that the 5-year overall survival was similar between groups [34].

### 2.6. Voice Quality

The post-TLM voice quality depends on the resection features [30]. Type I–III cordectomies are associated with adequate voice quality outcomes, while type IV–VI cordectomies may lead to significant dysphonia. In all cases, the voice quality outcomes improved from 6 to 12 months post-TLM with better improvements in young individuals compared with elderly patients [35]. The patients with dysphonia post-type IV-VI TLM may benefit from autologous fat medialization to improve the voice [36]. Speech therapy is another important postoperative way to improve the voice quality. Subjective and objective voice quality parameters may be used in the follow-up to measure the effectiveness of speech therapy [35].

## 3. Transoral Robotic Surgery

### 3.1. History of Robotic Surgery

Leonardo da Vinci (1452–1519) designed the first robot in 1495. The prototype consisted of a mechanical, armored knight used for entertaining the royalty [37]. The term “robot” originates from the Czech word “robota” meaning laborer, and it was first coined by the Czech writer Karel Capek (1890–1938) in his play Rossum Universal Robots in 1921. The most significant advances in robotic technology were the results of the research conducted by the U.S. Military and National Aeronautics and Space Administration (NASA) in the 1970s and 1980s. The objective of this research program was to produce robots that allowed the realization of surgery on the battlefield or in submarines without requiring the presence of a surgeon. Many surgical robots were developed since 1985. In 2001, the ZEUS robot was firstly used to perform a telerobotic transatlantic robot-assisted cholecystectomy with a surgeon located in Strasbourg and the patient located in New York (NY, USA) [38]. In 2003, Intuitive Surgical Inc. (Intuitive Inc., Sunnyvale, CA, USA) acquired Computer Motion Inc. and replaced the ZEUS^®^ robot by the *Da Vinci*^®^ surgical robot. Since 2003, the *Da Vinci* robot was spread worldwide, especially in urology, digestive surgery, and gynecology departments and became the most used robotic system used with over 6000 commercialized robots worldwide.

### 3.2. Concept

The *da Vinci* system robotic-assisted surgery was particularly used in urology and gynecology [39]. The robot system provides to the surgeon a 3D view of the surgical field, an important possibility of instrument moving, tremor removal, and an improvement of some movements in narrow spaces. The first transoral robotic surgery (TORS) for laryngeal (supraglottic) cancer was carried out in 2007 and reported adequate safety and feasibility outcomes [40,41]. At this time, the term ‘Transoral robotic surgery’ (TORS) was introduced in otolaryngology to describe this kind of procedure. In 2009, the Food and Drug Administration approved TORS procedures for cT1 or cT2 oral, oropharyngeal, or laryngeal carcinoma, as well as for benign tumors of the same locations. To date, head and neck surgery represents only 3% of the robotic market [37].

### 3.3. Indications and Contraindications

Throughout the last decade, TORS evolved from a proof-of-concept to a standard-of-care approach in centers with high-volume robotic surgeries. However, the current system is not adapted to all types of head and neck surgeries. To date, the commonly accepted indications for TORS remain cT1-T2 and some selected cT3-T4a of oropharynx and supraglottic regions in which the surgeon has adequate instrument-optic view triangulation. The interest of robotic approach in vocal fold tumors is not demonstrated regarding the robotic arm length that does not allow the realization of adequate vocal cord resection.

#### 3.3.1. TORS Supraglottic Laryngectomy

Since the first cases of TORS supraglottic laryngectomy [40,41], the approach was spread worldwide [42]. The European Laryngological Society classification can also be used for supraglottic laryngectomy TORS [13]. According to a recent systematic review, TORS supraglottic laryngectomy reported similar oncological outcomes to those of open procedures and is mainly carried out for the following indications: cT1 (35.8%), cT2 (48.6%), and some cT3 supraglottic cancers (13.9%) [42]. Precisely, the 24-month local and regional control rates ranged from 94.3% to 100% and 87.5% to 94.0%, respectively. The 2-year and 5-year overall survival rates ranged from 66.7% to 88.0% and 78.7% to 80.2%, respectively [42]. Importantly, the use of robotic approach for supraglottic cancers was associated with adequate postoperative functional outcomes [42,43]. In a large case series, Hans et al. reported that most patients (92%) re-started oral diet within the 24–48-h post-surgery, while transient tracheotomy was performed in 8% of patients [43]. The main complications include postoperative hemorrhages, aspiration, and pneumonia, but the complication rates were not higher than those of open supraglottic laryngectomy studies [43]. There is no significant influence of the tumor location (anterior and medial vs. lateral and piriform sinus) on the oncological outcomes [43].

#### 3.3.2. TORS-Cordectomy and Total Laryngectomy

TLM remains the standard surgical approach for the treatment of early glottic carcinoma. The feasibility of TORS in glottic surgery was first demonstrated by O’Malley et al. in 2006 in a canine model [44]. However, the use of TORS approach in human patients with glottic cancer is still controversial. In a recent review, we reported that TORS-cordectomy was associated with substantial exposure difficulties of glottic plan, inadequate hospital stay duration, additional costs related to TORS use and hospital stay, higher rates of feeding tube, tracheotomy, and complications [45]. The primary exposure difficulties were related to base of tongue hypertrophy and both size and rigidity of the robot arms that did not allow an adequate resection [45,46,47]. To date, although the lack of controlled studies, the usefulness of TORS-cordectomy with the current *Da Vinci* system (e.g., X, Xi) is not demonstrated.

As for cordectomy, the place of TORS for total laryngectomy is not yet well-defined. The first TORS total laryngectomy was reported in 2013 in a Franco-American cadaver study using the *da Vinci* System [48]. At that time, authors concluded that this approach was especially appropriate for patients undergoing salvage laryngectomy after radiation failure. Since then, only four small cohort studies including few cases were conducted, which preclude drawing any clear conclusion [49,50,51,52]. Overall, TORS total laryngectomy appears to be safe and effective, but its exact indications and limitations are not yet defined. Note that the procedure takes time, which is probably its primary limitation. The voice rehabilitation of patients benefiting from TORS total laryngectomy could be comparable with the postoperative outcomes of open total laryngectomy, but future controlled studies are needed [52].

## 4. Conclusions

The development of transoral laser microsurgery (TLM) was an important step in the improvement of the partial laryngeal surgery techniques. Although similar or better oncological and functional outcomes than open partial laryngectomy, TLM reported many exposure, learning curve, or surgical view limitations. The development of transoral robotic surgery led to new insights, especially for cT1-T2 and some cT3 oropharyngeal and supraglottic carcinomas. The future development of a new robotic system with smaller arms and instruments will be undoubtedly associated with new indications in the head and neck surgery and better oncological, surgical, and functional outcomes.

## Figures and Tables

**Figure 1 jcm-11-05769-f001:**
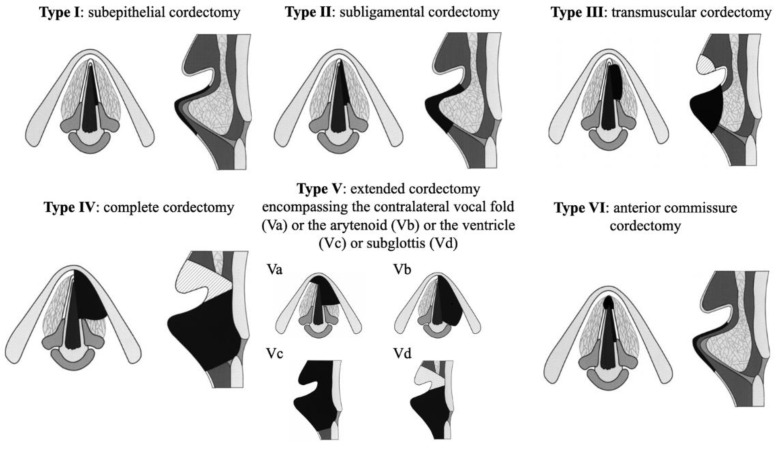
The European Laryngological Society classification of endoscopic cordectomies. This classification was published by Remacle et al., 2000 [9] and Remacle et al., 2007 [10].

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
