# Peer review of "Laryngeal Cancer Surgery: History and Current Indications of Transoral Laser Microsurgery and Transoral Robotic Surgery"

_jcm, 2022, doi:10.3390/jcm11195769_

Round 1
Reviewer 1 Report
Thanks for the invitation to review this manuscript. This is a well written review, discussing in a comprehensive manner the history and indications of modern conservative laryngeal surgery.
However, I think some changes and clarifications should be done.
1) On line 129, Authors wrote “129 These advantages supported the fact that 96% of patients themselves prefer being treated by TLM rather than radiotherapy [25].”, but perhaps it could be more correct to cite also the “opposite” references or other point of view, as well. Consider for example the following papers:
doi: 10.2147/OTT.S137210
doi: 10.4143/crt.2016.503
https://doi.org/10.1177/0145561319876905
https://doi.org/10.3389/fonc.2020.01669
2) Then, in line 165 “It is broadly accepted that a proportion of patients with positive margins do not develop recurrence, and the relationship between resection margin status and recurrence rate is still unclear [30].”
Please can you explain better the first part of the sentence to avoid contradictions also with the content of the paper cited in the reference 30?
3) Moreover, considering the main title, I would ask if it was really necessary to insert a paragraph about TORS oropharyngectomy (3.3.1),
4) Please correct minor edits, such as “line 82 cricoarytenoid join”: joint
5) Maybe there are many self-citations (13 on 61, about 18% of all citations). Although not inappropriate or extreme, I think a reduction could be more acceptable.
Author Response
Reviewer 1
Thanks for the invitation to review this manuscript. This is a well written review, discussing in a comprehensive manner the history and indications of modern conservative laryngeal surgery.
However, I think some changes and clarifications should be done.
1) On line 129, Authors wrote “129 These advantages supported the fact that 96% of patients themselves prefer being treated by TLM rather than radiotherapy [25].”, but perhaps it could be more correct to cite also the “opposite” references or other point of view, as well. Consider for example the following papers:
doi: 10.2147/OTT.S137210
doi: 10.4143/crt.2016.503
https://doi.org/10.1177/0145561319876905
https://doi.org/10.3389/fonc.2020.01669
Answer:
We thank the reviewer for these comments.
We agree there is a debate between TLM and radiotherapy in the treatment of T1-T2 glottic cancers. However, the purpose of this article is to show the evolution of surgical treatments for laryngeal cancer: from partial laryngectomies (part I) to minimally invasive surgery. A single study is insufficient to discuss all the existent data regarding the differences between TLM/TORS versus radiation therapy. Consequently, we changed the title of the article:
“Laryngeal Cancer Surgery: History and Current Indications of Transoral Laser Microsurgery and Transoral Robotic Surgery”
These advantages supported the fact that 96% of patients themselves prefer being treated by TLM rather than radiotherapy [25].”,
The aim of this study was to investigate treatment preferences in patients with early glottic carcinoma who were given a choice between TLM and RT. Patients were informed concerning each treatment by a laryngologist and a radiotherapist. Of 175 patients, 96% chossed TLM, and 7 patients choose RT. The decision was based mainly on treatment duration and the wider treatment options in case of recurrence.
2) Then, in line 165 “It is broadly accepted that a proportion of patients with positive margins do not develop recurrence, and the relationship between resection margin status and recurrence rate is still unclear [30].”
Please can you explain better the first part of the sentence to avoid contradictions also with the content of the paper cited in the reference 30?
Answer:
We thank the reviewer for these comments.
The scientific literature and the given reference (30) sustain that patients with positive margins do not develop recurrence.
We changed the text in order to clarify the message and avoid contradictions.
“It is broadly accepted that a proportion of patients with positive margins do not develop recurrence, and the relationship between resection margin status and recurrence rate is still unclear [30]. The pathological examination of post-TLM specimen is a challenging task following laser surgery inducing difficulties in margin status outcome. In case of suspected positive margins, 3 attitudes are reported in the literature. First, the patient will follow a TLM re-intervention or radiotherapy. Second, a second endoscopic evaluation 6 to 8 weeks after the TLM may be performed to determine the need of reintervention. Third, the surgeon can propose a follow-up approach and a particular attention to voice quality impairment or videolaryngostroboscopy suspicion of recurrence/progression during close follow-up.”
3) Moreover, considering the main title, I would ask if it was really necessary to insert a paragraph about TORS oropharyngectomy (3.3.1),
Answer:
We thank the reviewer for these comments.
We removed the TORS oropharyngectomy paragraph.
4) Please correct minor edits, such as “line 82 cricoarytenoid join”: joint
Answer:
We thank the reviewer for these comments. It's corrected.
5) Maybe there are many self-citations (13 on 61, about 18% of all citations). Although not inappropriate or extreme, I think a reduction could be more acceptable.
Reviewer 2 Report
· In my opinion, you couldn’t compare TLM with TORS since the former is used mainly for glottic lesions and the latter for oropharyngeal and supraglottic lesions, as you described.
· You made a detailed description of advantages and limits of TLM; however, you wrote about TORS mainly in terms of indications and advantages, less about limits and drawbacks. Indeed, also TORS has some limitations: duration of surgery that is longer than TLM, higher costs, its unavailability in most hospitals.
· To make a comparison, you should analyze the same issues for both surgeries: indications and limitations, barriers, benefits, functional and oncological outcomes, duration of surgery, costs… As suggestion, you could analyze TLM and TORS for the same indication: supraglottic and glottic surgeries. In this case, you may do a comparison between the two surgical approaches. TORS allows a better exposition but the same issues of TLM occur: management of anterior commissure and paraglottic space. They are both transoral surgeries and so have the same limits.
· Conclusion section: the conclusion, as well as the entire text, assumes that TLM and TORS are comparable but, as you wrote, they have different indications.
· References 23 and 24 are the same. There are a lot of self-citations, you should reduce them.
Author Response
In my opinion, you couldn’t compare TLM with TORS since the former is used mainly for glottic lesions and the latter for oropharyngeal and supraglottic lesions, as you described.
- You made a detailed description of advantages and limits of TLM; however, you wrote about TORS mainly in terms of indications and advantages, less about limits and drawbacks. Indeed, also TORS has some limitations: duration of surgery that is longer than TLM, higher costs, its unavailability in most hospitals.
- To make a comparison, you should analyze the same issues for both surgeries: indications and limitations, barriers, benefits, functional and oncological outcomes, duration of surgery, costs… As suggestion, you could analyze TLM and TORS for the same indication: supraglottic and glottic surgeries. In this case, you may do a comparison between the two surgical approaches. TORS allows a better exposition but the same issues of TLM occur: management of anterior commissure and paraglottic space. They are both transoral surgeries and so have the same limits.
- Conclusion section: the conclusion, as well as the entire text, assumes that TLM and TORS are comparable but, as you wrote, they have different indications.
- References 23 and 24 are the same. There are a lot of self-citations, you should reduce them.
We thank the reviewer for these comments.
The aim of our study was to describe the conservative treatment evolution in laryngeal cancers, from open partial laryngectomies to minimally invasive laryngeal surgery. Therefore, we discuss the indications and limitations while using TLM/TORS and we don’t compare the two surgical techniques. Scientific literature does not provide enough data to compare TLM and TORS especially in supraglottic cancers.
Reviewer 3 Report
Dear Authors,
I read Your review about the history of the development of mini-invasive approaches to the larynx with interest. I have only a few comments, which are detailed below:
- I would consider to mention also the endoscopic supraglottic laryngectomy classification by the European Laryngological Society, in addition to the classification of cordectomies;
- Paragraph 3.3.1 concerning TORS oropharyngectomy is out of the scope of the paper, I would consider to remove it;
- I think the manuscript would benefit from English language revision by a mother-tongue speaker with an expertise in clinical and scientific writing.
Author Response
Dear Authors,
I read Your review about the history of the development of mini-invasive approaches to the larynx with interest. I have only a few comments, which are detailed below:
- I would consider to mention also the endoscopic supraglottic laryngectomy classification by the European Laryngological Society, in addition to the classification of cordectomies;
Answer:
We thank the reviewer for these comments.
We have reported the ELS classification for supraglottic laryngectomy TLM or TORS.
Remacle M, Hantzakos A, Eckel H, Evrard AS, Bradley PJ, Chevalier D, Djukic V, de Vincentiis M, Friedrich G, Olofsson J, Peretti G, Quer M, Werner J. Endoscopic supraglottic laryngectomy: a proposal for a classification by the working committee on nomenclature, European Laryngological Society. Eur Arch Otorhinolaryngol. 2009 Jul;266(7):993-8.
- Paragraph 3.3.1 concerning TORS oropharyngectomy is out of the scope of the paper, I would consider to remove it;
Answer:
We thank the reviewer for these comments.
We removed the TORS oropharyngectomy paragraph.
- I think the manuscript would benefit from English language revision by a mother-tongue speaker with an expertise in clinical and scientific writing.
